# Antioxidant and Antihyperglycemic Effects of *Ephedra foeminea* Aqueous Extract in Streptozotocin-Induced Diabetic Rats

**DOI:** 10.3390/nu14112338

**Published:** 2022-06-02

**Authors:** Maha N. Abu Hajleh, Khaled M. Khleifat, Moath Alqaraleh, Esra’a Al-Hraishat, Muhamad O. Al-limoun, Haitham Qaralleh, Emad A. S. Al-Dujaili

**Affiliations:** 1Department of Cosmetic Science, Pharmacological and Diagnostic Research Center, Faculty of Allied Medical Sciences, Al-Ahliyya Amman University, Amman 19328, Jordan; m.abuhajleh@ammanu.edu.jo; 2Faculty of Allied Medical Sciences, Al-Ahliyya Amman University, Amman 19328, Jordan; alkhkha@mutah.edu.jo; 3Department of Biological Sciences, Mutah University, Mutah 61710, Jordan; 620190305017@mutah.edu.jo (E.A.-H.); moallimoun@mutah.edu.jo (M.O.A.-l.); 4Department of Medical Laboratory Sciences, Faculty of Science, Mutah University, Mutah 61710, Jordan; haitham@mutah.edu.jo; 5Pharmacological and Diagnostic Research Center, Faculty of Pharmacy, Al-Ahliyya Amman University, Amman 19328, Jordan; m.alqaraleh@ammanu.edu.jo; 6Centre for Cardiovascular Science, Queen’s Medical Research Institute, University of Edinburgh, Edinburgh EH16 4JT, UK

**Keywords:** *Ephedra foeminea*, antioxidant, antidiabetic, glutathione peroxidase, interleukin 1beta, LC-MS

## Abstract

Background: *Ephedra foeminea* is known in Jordan as Alanda and traditionally. It is used to treat respiratory symptoms such as asthma and skin rashes as an infusion in boiling water. The purpose of this study was to determine the antidiabetic property of *Ephedra foeminea* aqueous extract in streptozotocin-induced diabetic rats. Methods: The aqueous extract of *Ephedra foeminea* plant was used to determine the potential of its efficacy in the treatment of diabetes, and this extract was tested on diabetic rats as a model. The chemical composition of *Ephedra foeminea* aqueous extract was determined using liquid chromatography–mass spectrometry (LC-MS). Antioxidant activity was assessed using two classical assays (ABTS and DPPH). Results: The most abundant compounds in the *Ephedra foeminea* extract were limonene (6.3%), kaempferol (6.2%), stearic acid (5.9%), β-sitosterol (5.5%), thiamine (4.1%), riboflavin (3.1%), naringenin (2.8%), kaempferol-3-rhamnoside (2.3%), quercetin (2.2%), and ferulic acid (2.0%). The antioxidant activity of *Ephedra foeminea* aqueous extract was remarkable, as evidenced by radical scavenging capacities of 12.28 mg Trolox/g in ABTS and 72.8 mg GAE/g in DPPH. In comparison to control, induced diabetic rats treated with *Ephedra foeminea* extract showed significant improvement in blood glucose levels, lipid profile, liver, and kidney functions. Interleukin 1 and glutathione peroxidase levels in the spleen, pancreas, kidney, and liver of induced diabetic rats treated with *Ephedra foeminea* extract were significantly lower than in untreated diabetic rats. Conclusions: *Ephedra foeminea* aqueous extract appears to protect diabetic rats against oxidative stress and improve blood parameters. In addition, it has antioxidant properties that might be very beneficial medicinally.

## 1. Introduction

Diabetes mellitus is a metabolic disease that remains a serious public health problem on a global scale. The use of insulin and various other drugs in controlling diabetes is widespread, and diabetic complications such as neuropathy, vascular dysfunction, nephropathy, and retinopathy are common [1,2]. It is now well established that oxidative damage plays a role in the etiology of diabetes and the development of diabetic complications [3]. The search for a new antidiabetic agent with antioxidant properties is highly recommended.

Plant extracts and natural medicinal therapies are being used to treat a variety of diseases, one of which is diabetes. In fact, over 1000 medicinal plants are traditionally used to treat diabetes, and only 350 have been recorded as antidiabetic agents [4,5]. Natural products used in traditional medicine have a significant influence on the treatment of diabetes mellitus, and plant extracts with antioxidant function have been reported to possess marked antidiabetic properties. They are rich in antioxidant compounds such as flavonoids, tannins, phenolic acids, and alkaloids that enhance pancreatic cell efficiency and increase insulin levels by decreasing glucose uptake through the intestinal wall [6,7]. Numerous studies have demonstrated that antioxidants improve oxidative stress and lipid peroxidation markers in diabetic subjects. In streptozotocin-induced diabetic rats, Andallua and Varadacharyulub [8] discovered that Morus indica extract has antioxidant properties. Another study demonstrated the antioxidant activity of Viscum album in the heart and kidney tissues of diabetic rats [9].

*Ephedra foeminea* is known in Jordan as Alanda and traditionally. It is used to treat respiratory symptoms such as asthma and skin rashes [10]. The plant is a small evergreen climbing or hanging shrub and found in the Eastern Mediterranean region. Despite the traditional use of aerial parts of *Ephedra foeminea* as a hypoglycemic agent, no attempts have been made to study the antidiabetic property of any *Ephedra foeminea* extract. Therefore, this study evaluated the antidiabetic activity of aerial parts of *Ephedra foeminea* aqueous extract in Streptozotocin-Induced Diabetic rats.

A range of bioactive compounds have been known in Ephedra sp., such as polyphenols and alkaloids [11,12]. All species of the Ephedra genus belong to the chemotype known as the ephedrine chemotype, except *Ephedra foeminea*, which lacks this type of alkaloid [13].

The aims of this study were to evaluate the in vivo antidiabetic property of *Ephedra foeminea* aqueous extract, to determine its antioxidant activity, and to investigate the antioxidant protective effect of this plant extract by measuring the expression levels of interleukin 1beta and glutathione peroxidase. The chemical composition of *Ephedra foeminea* aqueous extract will also be characterized using liquid chromatography–mass spectrometry (LC-MS).

## 2. Material and Methods

### 2.1. Plant Materials

*Ephedra foeminea* aerial parts were collected in May 2020 from the Dana district of Tafilah, south of Jordan. The plant was identified using Flora Palaestina, Part 3 as a guide [14], and further confirmed by Prof. Saleh Al-Quran (Department of Biology, Mutah University, Jordan). A voucher specimen (E1/2020) was deposited in the Department of Biology, Mutah University, Jordan. The collected plant materials were air-dried at room temperature in the shade, processed into a fine powder, and stored in aliquots at 4 °C.

### 2.2. Aqueous Extract

The plant material was grounded, and the aqueous extract was prepared using distilled water according to Jaiswal et al. [15] with some modifications. This was performed by soaking 50 g of *Ephedra foeminea* shoots in 500 mL of distilled water, and the mixture was incubated at room temperature for 24 h. The water was then removed by filtration, collected, and immediately frozen at −20 °C. The frozen samples were freeze-dried, and the produced dried materials were collected and stored in aliquots at 4 °C.

### 2.3. Determination of Chemical Composition Using LC-MS

HPLC separation of *Ephedra foeminea* aqueous extract was performed with the mobile phase containing solvent A and B in a gradient, where A was 0.1% (*v*/*v*) formic acid in water, and B was 0.1% (*v*/*v*) formic acid in acetonitrile for the following gradients: 5% B for 5 min, 5–100% B for 15 min, and 100% for 5 min at a flow rate of 0.5 mL/min. The column used was the Agilent Zorbax Eclipse XDB-C18 (2.1 × 150 mm × 3.5 µm). The oven temperature used was 25 °C, and the sample injection volume was 1 μL (18 mg/mL in methanol). The eluent was monitored by a Shimadzu LC-MS 8030 with an electrospray ion mass spectrometer (ESI-MS) in positive ion mode and scanned from 100 to 1000 m/z. ESI was conducted by using a fragmentor voltage of 125 V and a skimmer of 65 V. High-purity nitrogen (99.999%) was used as a drying gas at a flow rate of 10 L/min, a nebulizer at 45 psi, and a capillary temperature of 350 °C. In parallel, 0.1% formic acid was used as a blank. A sample was injected into the mass detector using a Shimadzu CBM-20A system controller, an LC-30AD pump, an SIL-30AC autosampler with a cooler, and a CTO-30 column oven [16].

### 2.4. Antioxidant Activity

#### 2.4.1. ABTS Free Radical Scavenging Assay

The ABTS test was used to determine the scavenging activity of the *Ephedra foeminea* extract [17]. In this experiment, 7 mM of ABTS radical in 2.15 mM of potassium persulfate was prepared and incubated in the dark at room temperature for 16 h. Then, the mixture was diluted with ethanol to achieve an absorbance of 0.70 ± 0.2 at 734 nm. To determine the ABTS radical scavenging ability, 20 µL of *Ephedra foeminea* extract was added to 2.0 mL of ABTS solution. Positive and negative controls were also prepared in the same way. The prepared solutions were incubated for 6 min, and the absorbance at 734 nm was measured. The ABTS scavenging activity was estimated using the formula:ABTS radical scavenging activity (%) = [(A0 − A1)/A0 × 100]
where A0: absorbance for control and A1: absorbance for sample. Trolox was applied to prepare a standard curve at concentrations ranging from 0 to 600 mg/mL. The activity of ABTS as a radical scavenger is represented in mg Trolox equivalents (TE)/gm extracts [18].

#### 2.4.2. DPPH Radical Scavenging Activity

The antioxidant activity of the *Ephedra foeminea* extract was also determined using the DPPH assay [19]. A volume of 1.9 mL of 0.1 mM of 1,1-diphenyl-2-picryl-hydrazyl (DPPH) in methanol was mixed with 0.1 mL of *Ephedra foeminea* extract. After vigorous shaking, the mixture was incubated at room temperature for 30 min and tested for absorbance at 517 nm. As a positive control, gallic acid was utilized. The following formula is used to compute the DPPH radical scavenging activity:DPPH radical scavenging activity (%) = [(A0 − A1)/A0 × 100]
where A0: absorbance of control and A1: absorbance of test sample [18,20].

### 2.5. Antidiabetic Activity

#### 2.5.1. Animals

Albino male Wister rats (200–250 g) were obtained from Jordan University’s animal house, Amman, Jordan. Animals were fed a standard laboratory pellet diet and were given tap water to drink. This research work was approved by the institutional animal ethics committee (Decision Number 2072021), in accordance with Mutah University’s animal guidelines. The body weight of rats was recorded at baseline and weekly.

#### 2.5.2. Induction of Diabetes

The induction of diabetes was performed using streptozotocin (STZ). The fasted animals were intraperitoneally injected with STZ (55 mg/kg body weight) prepared in 0.1 M saline solution (pH 4.5), while the control group was injected with citrate buffer alone. To keep the citrate buffer at a lower temperature, ice was maintained around the glass beaker containing it. During the 24 h, the STZ-injected animals were given a 5% glucose solution to elevate the blood glucose level that could be produced by STZ. After 4 days, the blood glucose level was measured, and animals with a blood glucose level of above 300 mg/dL were considered diabetic [21].

#### 2.5.3. Grouping of Animals and Experimental Procedure

Thirty rats were used in total, divided into five groups of six rats each. These groups were organized as follows:Group 1 (G1): Normal, nondiabetic rats were given citrate buffer on Day 1 (on the same day of STZ injection for the STZ-induced diabetic rats’ groups) and normal saline solution orally starting from Day 4.Group 2 (G2): Diabetic rats were given normal saline solution orally.Group 3 (G3): Diabetic rats were given metformin (100 mg/Kg) orally starting from Day 4.Group 4 (G4): Diabetic rats were given *Ephedra foeminea* extract (100 mg/Kg) orally starting from Day 4.Group 5 (G5): Diabetic rats were given *Ephedra foeminea* extract (100 mg/Kg) orally from Day 1 (on the same day of STZ injection) up to Day 4. The oral LD50 of Ephedra aqueous extract was reported to be in the range of 4000–8000 mg/kg [22].

After 4 weeks of treatment, blood samples were taken to evaluate the lipid profile, liver, and kidney functions. On the same day, the rats were anesthetized under general anesthesia induced by diethyl ether inhalation as described by Alsarayreh et al. [23], and their organs, including liver, spleen, pancreas, and kidney, were collected and preserved in phosphate-buffered saline (PBS) (pH 7.4) at −80 °C for further use. The body weight of all rats was recorded on Day 1 and thereafter once weekly.

### 2.6. Analysis of Blood Parameters

Blood glucose level was measured using a glucometer (Glucolab, Infopia Co., Ltd., Anyang 431-080, Gyeonggi-do, Korea) Serum levels of cholesterol, triglyceride, low-density lipoprotein (LDL), high-density lipoprotein (HDL), alanine aminotransferase (ALT), aspartate aminotransferase (AST), alkaline phosphatase (ALKP), total bilirubin, total protein, creatinine, urea, phosphorus, chloride ion, and sodium ion were all determined using chemistry analyzers (Abbott Alinity Series).

#### 2.6.1. Interleukin 1beta Measurement

The cytokine, interleukin 1beta, level was determined in the liver, kidney, spleen, and pancreas for all groups. Tissue samples were allowed to thaw, mixed with PBS (pH 7.4), and homogenized. The homogenized samples were mixed with 1.0 mL of lysis buffer and incubated at room temperature for 30 min with gentle agitation. The samples were then centrifuged at 3000 rpm for 20 min at 4 °C, and the supernatants were collected. The interleukin 1beta level was measured in these tissue samples using the interleukin 1beta ELISA kit (Abcam, Cambridge, MA, USA) according to the manufacturer’s instructions.

#### 2.6.2. Glutathione Peroxidase Levels

The antioxidant enzyme, glutathione peroxidase, level was determined in the liver, kidney, spleen, and pancreas for all groups. Frozen tissue samples were allowed to thaw, mixed with PBS (pH 7.4), and then homogenized. The samples were centrifuged at 3000 RPM at 4 °C for 20 min, and the supernatants were collected. The level of glutathione peroxidase was measured in the tissue samples using the glutathione peroxidase kit (Shanghai Coon Koon Biotech Co., Ltd., Shanghai, China) according to the manufacturer’s instructions.

### 2.7. Statistical Analysis

For all the experiments performed, the means and standard deviations of six independent tests were calculated. SPSS version 22 was used to analyze the data (SPSS Inc., Chicago, IL, USA). The significant differences were figured using one-way analysis of variance (ANOVA), followed by Dunnett’s post hoc test. Statistical significance was defined as a value of *p* < 0.05 (indicated by *), <0.001 (indicated by **) or <0.0001 (indicated by ***).

## 3. Results

### 3.1. Chemical Composition of Ephedra foeminea Aqueous Extract

A total of fifty compounds were identified, representing 94.96% of the total identified compounds (Figure 1). About 32.36% of these compounds were identified as flavonoids and phenolics (Table 1) whereas 63.9% were identified as terpenes, organic acids, and others (Table 2). The most dominant compounds in the *Ephedra foeminea* extract reported here were limonene (6.3%), kaempferol (6.2%), stearic acid (5.9%), β-sitosterol (5.5%), thiamine (4.1%), riboflavin (3.1%), naringenin (2.8%), kaempferol-3-rhamnoside (2.3%), quercetin (2.2%), and ferulic acid (2.0%).

### 3.2. Antioxidant Activity

The antioxidant properties of the aqueous extract of *Ephedra foeminea* were determined using two methods: ABTS and DPPH. The antioxidant activity against ABTS radical scavenging was determined in terms of Trolox equivalents. *Ephedra foeminea* aqueous extract had a scavenging capacity of 12.28 mg Trolox/g plant extract for the radical ABTS. The total antioxidant activity as determined by DPPH was comparable to that determined by ABTS, with an observed value of 72.8 mg GAE/g plant extract. At a low concentration (0.4 mg/mL), *Ephedra foeminea* aqueous extract had a DPPH scavenging potential of more than 50%.

### 3.3. Antidiabetic Activity

#### 3.3.1. Blood Glucose Levels

The protective effect of *Ephedra foeminea* extract was proposed in this study based on the reduction in blood glucose of induced diabetic rats treated with *Ephedra foeminea* extract on the first day of the experiment (the same day of the STZ injection) and for four days only (short-term treatment) (G5). The reduction in blood glucose levels in rats treated with *Ephedra foeminea* extract on the fourth day of the experiment (long-term treatment) was proposed to investigate the curative effect of *Ephedra foeminea* extract (G4). Each group’s blood glucose level was determined weekly for a period of four weeks (Figure 2). A significant gradual increase in blood glucose concentrations (4-fold) was reported in untreated induced diabetic rats (G2), while a significant gradual decrease in blood glucose concentrations was reported in induced diabetic rats treated with metformin (G3) and *Ephedra foeminea* extract (G4 and G5). Untreated induced diabetic rats (G2) had significantly higher glucose levels (4-fold) than the normal group (G1) (*p* < 0.05). In addition, the glucose concentration was significantly decreased in induced diabetic rats treated with metformin (G3) and *Ephedra foeminea* extract (G4) compared to the untreated induced diabetic rats (G2). Although a euglycemic effect was observed in induced diabetic rats administered short-term treatment with *Ephedra foeminea* extract (G5), the concentration of blood glucose in this group was higher than that of the long-term treatment group (G4).

#### 3.3.2. Body Weight

Each group’s body weight was determined weekly for a period of four weeks (Figure 3). Nontreated induced diabetic rats (G2) showed a significant decrease in body weight at the end of the experiment compared to the first day of the experiment (*p* < 0.05). From Day 15, body weights were significantly (*p* < 0.0001) increased for diabetic rats treated with metformin (G3) and *Ephedra foeminea* extract (G4 and G5) compared to the untreated induced diabetic rats.

### 3.4. Serum Lipid Profile (Cholesterol, Triglyceride, LDL, and HDL)

The effect of *Ephedra foeminea* extract on the blood concentrations of total cholesterol, triglycerides, low-density lipoprotein (LDL), and high-density lipoprotein (HDL) in the tested groups was studied. As shown in Figure 4, nontreated induced diabetic rats (G2) had a significant rise in cholesterol, triglycerides, and LDL concentrations and a nonsignificant boost in HDL concentrations compared with the non-(G1) group (*p* > 0.05). However, the levels of triglycerides, cholesterol, and LDL were significantly reduced in induced diabetic rats treated with metformin (G3) and *Ephedra foeminea* extract (G4 and G5) compared with the untreated induced diabetic rats (G2) (P0.05). In all experimental groups, there was no significant difference in HDL concentration.

### 3.5. Serum Markers of Liver Function (ALT, AST, ALKP), Total Bilirubin, and Total Proteins

Liver enzymes (ALT, AST, ALKP), total bilirubin, and total protein levels in the rats’ serum samples were determined to evaluate the functions of the liver. As shown in Figure 5, untreated induced diabetic rats (G2) showed a significant increase in the concentrations of ALT, AST, ALKP, total bilirubin, and total protein compared to the normal group (*p* < 0.05), indicating hepatocellular damage associated with diabetes [24,25]. There was a significant decrease in the liver biomarker concentrations in the induced diabetic rats treated with metformin (G3) and *Ephedra foeminea* extract (G4) compared to the untreated induced diabetic rats (G2). Notably, the short-term treatment of induced diabetic rats with *Ephedra foeminea* extract (G5) was significantly higher than that of long-term treatment with *Ephedra foeminea* extract (G4).

### 3.6. Evaluation of Kidney Function Test Biomarkers

The kidneys eliminate waste from the body, thereby maintaining the ideal concentrations of urea, uric acid, creatinine, and ions. However, these metabolites accumulate in the bloodstream during renal diseases or renal damage that is commonly caused by chronic hypertension and diabetes [26]. The levels of these kidney parameters have been determined to assess the kidney’s functions. As shown in Figure 6, untreated induced diabetic rats (G2) showed a significant increase in the concentrations of creatinine, urea, phosphorus, chloride, and sodium ions compared to the normal group (*p* < 0.05), indicating renal damage. There was a significant decrease in these kidney marker concentrations in the induced diabetic rats treated with metformin (G3) and *Ephedra foeminea* extract (G4) with long-term treatment, indicating that the extract of *Ephedra foeminea* may elevate kidney damage and function. This elevation in kidney function markers in induced diabetic rats treated with *Ephedra foeminea* extract (G5) appears to be lower than that in induced diabetic rats treated with *Ephedra foeminea* extract (G4).

### 3.7. Interleukin 1beta Level

Interleukin 1 is produced early and could be used in the circulation as a biomarker of type1 DM risk. The concentration of the proinflammatory cytokine interleukin 1beta was determined in the spleen, pancreas, kidney, and liver of all tested groups in this study. Figure 7 shows that the concentration of interleukin 1beta increased significantly in untreated induced diabetic rats (G2) compared to the control group (G1). In contrast, a significant decrease in the level of interleukin 1beta was shown in induced diabetic rats treated with metformin (G3) and *Ephedra foeminea* extract (G4 and G5). Significant reductions in interleukin 1beta levels after administration of *Ephedra foeminea* extract are attributed to *Ephedra foeminea’s* hepatoprotective properties, which are attributed in part to its ability to inhibit proinflammatory cytokines such as interleukin 1beta.

### 3.8. Glutathione Peroxidase Level

As shown in Figure 8, the concentration of glutathione peroxidase is significantly increased in untreated induced diabetic rats (G2) compared to the normal group (G1), suggesting acute oxidative stress. However, when metformin (G3) and *Ephedra foeminea* extract (G4 and G5) were administered, glutathione peroxidase levels decreased significantly. As a result, the antioxidant enzyme status is restored to normal levels. In addition, the level of glutathione peroxidase was higher in the livers of induced diabetic rats administered short-term treatment of *Ephedra foeminea* extract (G5), indicating that a short-term treatment may not efficiently treat the liver injury.

## 4. Discussion

*Ephedra foeminea* plant is widely used in traditional therapy of many diseases in Jordan, where it is boiled in water for a few minutes depending on the user, and some people soak it in boiling water for at least two hours. As a result, we decided to use the aqueous extract of this plant to determine the extent of its antidiabetic property in a diabetic rat’s model. We have observed that the antioxidant capacity potency (DPPH, ABTS, and FRAP) of water-based extracts were comparable to those obtained after the long-period extractions (>5 h) using organic solvents [27]. The results of this study appeared to partially agree with those reported by Ibragic and Sofić [28], who found that *Ephedra foeminea* is rich in phenolics and flavonoids. While previous LC-MS analysis revealed that it contains prunin, quercetin, rutin, vitexin, hyperoside, and isoorientin [29], the most dominant compounds in our *Ephedra foeminea* extract reported here were limonene, kaempferol, stearic acid, β-sitosterol, thiamine, riboflavin, naringenin, kaempferol-3-rhamnoside, quercetin, and ferulic acid. However, it is the only Ephedra species that lacks the ephedrine alkaloid, and the quinoline alkaloids 6-hydroxykynurenic acid has been isolated as the major alkaloid in *Ephedra foeminea* [13]. The content of extracts from a particular plant species might vary according to time of collection, mode of extraction, geographical origin, and vegetative state [30]. The total antioxidant activity as determined by DPPH was comparable to that determined by ABTS, with an observed value of 72.8 mg GAE/g plant extract. At a low concentration (0.4 mg/mL), *Ephedra foeminea* aqueous extract had a DPPH scavenging potential of more than 50%. It is worth mentioning that high-polar extracts, such as water, methanol, and butanol extracts, display stronger antioxidant activity compared to low-polar solvents [12,31,32]. The potential antioxidant activity of *Ephedra foeminea* extract appears to be linked to its content of flavonoids, terpenes, and vitamins and, to a greater extent, to the presence of high concentrations of limonene, kaempferol, and ferulic acid [33,34,35].

The significant increase in glucose levels reported in streptozotocin-injected rats could be due to STZ’s destruction of pancreatic beta cells, resulting in insulin deficiency [36]. However, the reduction in blood glucose levels of induced diabetic rats treated with metformin (G3) showed that metformin exhibited a euglycemic effect. Metformin is particularly known to be effective at lowering blood sugar levels in patients with type 2 diabetes. This is primarily accomplished by inhibiting hepatic gluconeogenesis and improving peripheral insulin sensitivity [37]. Subsequently, metformin may cause weight loss and may increase the risk of euglycemic effect [38]. The protective effect of *Ephedra foeminea* extract was proposed in this study based on the reduction in blood glucose of induced diabetic rats treated with *Ephedra foeminea* extract on the first day of the experiment and for four days only (short-term treatment) (G5). The reduction in blood glucose levels in rats treated with *Ephedra foeminea* extract on the fourth day of the experiment (long-term treatment) was proposed to investigate the curative effect of *Ephedra foeminea* extract (G4). Previous research has shown that Ephedra extracts and isolated compounds have an antidiabetic effect, and that ephedrans (A, B, C, D, and E 100 mg/Kg) extracted from the stem of *Ephedra distachya* have the potential to lower blood glucose [39]. Ephedrine, an alkaloid, has been revealed to mediate the recovery of chemically atrophied pancreatic islets, and it has been proposed that this alkaloid may be able to restore normal insulin secretion and blood glucose level control [40]. Moreover, an ephedrine-rich extract had an impressive euglycemic effect in an obese type 2 diabetes experimental model, which was linked to lower tumor necrosis factor (TNF-α) expression and higher peroxisome proliferator-activated receptor (PPAR-γ) expression [41,42]. Ephedrine and ephedrine derivatives demonstrated antidiabetic properties by inhibiting dipeptidyl peptidase IV (DPP-IV), which normally acts by increasing the level of insulin after meals [34,43,44]. Lee et al. [45] noticed that E. pachyclada and its main active ingredient, quinoline-2-carboxylic acid, have antidiabetic properties by inhibiting the enzymes α-glucosidase and α-amylase.

The weight loss in diabetic rats is thought to be due to the increased catabolism of fats and proteins used as a source of energy in the absence of carbohydrates. Insulin is critical for skeletal muscle protein synthesis and proteolysis regulation, and the observed effect is primarily due to insulin deficiency [46,47]. For healthy rats, there was no significant increase in body weight that could not be seen by the curve, which began at 239 g and increased to 247 g after 22 days before stabilizing at 30 days. However, no significant change in body weight was observed in the control groups (rats) when compared to streptozotocin-induced diabetic rats treated with aqueous extracts of other plants in another study [48,49]. Induced diabetic rats treated with metformin (G3) showed a significant increase in body weight. In contrast to our results metformin causes weight loss or decreases weight gain in type 2 diabetes and nondiabetic obesity patients [50,51,52,53] Interestingly, no significant change in body weight was reported for the induced diabetic rats treated with *Ephedra foeminea* extracts. However, no conclusive evidence that metformin significantly improves cholesterol, triglyceride, LDL, and HDL levels in type 1 diabetes patients [54]. The authors noticed that metformin reduced cholesterol and triglyceride levels, had no effect on HDL levels, and decreased LDL levels in type 1 diabetic patients in a meta-analysis study. In our study, daily administration of *Ephedra foeminea* extract normalized the lipid profile in diabetic animals for 30 days. However, an aqueous extract of *Ephedra sinica* found no effect on the serum lipid profile of normal rats [55]. *Ephedra sinica* was effective as an antihyperlipidemic agent and increased lipid metabolism by other researchers [56]. *Ephedra alata* extracts have been shown to increase lipase activity and stimulate lipid absorption, resulting in higher levels of cholesterol and LDL in the blood and increased body weight [57].

The levels of ALT, AST, ALKP, total bilirubin, and total proteins in the experimental rats’ serum were determined to evaluate liver functions. Untreated induced diabetic rats (G2) had a significant increase in all these parameters compared to the control group, indicating hepatocellular damage associated with diabetes [24,25]. Our study showed that there was a significant decrease in the concentrations of liver biomarkers in induced diabetic rats treated with metformin (G3) and *Ephedra foeminea* extract (G4) compared to untreated induced diabetic rats (G2). Notably, the short-term effect of *Ephedra foeminea* extract (G5) was significantly greater than the long-term effect of *Ephedra foeminea* extract (G4). The reduction in these liver biomarkers after treatment with *Ephedra foeminea* extract suggests that this extract mediates the repair of liver damage and restores its functions. Because the aqueous extract lacks ephedrine, the safety of using *Ephedra foeminea* extract appears to be linked to its chemical composition. Ephedrine has been linked to hepatocellular cell apoptosis and lowered liver function [58].

In renal diseases or renal damage, urea, uric acid, creatinine, and electrolytes accumulate in the bloodstream, which is frequently caused by chronic hypertension and diabetes [26]. Untreated induced diabetic rats (G2) had a significant increase in creatinine, urea, phosphorus, chloride, and sodium ion concentrations compared with the control group (*p* < 0.05), implying renal damage. Long-term treatment with metformin (G3) and *Ephedra foeminea* extract (G4) resulted in a significant reduction in these kidney markers, indicating that *Ephedra foeminea* extract may boost kidney damage and functions. This effect in induced diabetic rats treated with *Ephedra foeminea* extract (G5) appears to be lower than the increase in kidney function in induced diabetic rats treated for long-term treatment with *Ephedra foeminea* extract (G4). In cisplatin-induced nephrotoxicity mice, administration of *Ephedra alata* methanolic extract normalized relative kidney and body weight, restored biochemical and oxidative stress parameters, and decreased DNA damage and IFNγ levels [59].

It appears that the long-term treatment with *Ephedra foeminea* extract for 4 weeks improves blood glucose level, lipid profile, liver functions, and kidney functions better than the short-term treatment for 4 days. Interestingly, it was stated that maximal efficacy of plant-based therapy should be sustained for a longer length of time [60,61,62]. Proinflammatory cytokines such as interleukin 1 are potent inflammatory inducers that may play a role in the pathophysiological mechanisms of diabetes mellitus [63]. IL-1 is produced early and used as a biomarker of T1DM risk. The concentration of the proinflammatory cytokine interleukin 1beta was determined in the spleen, pancreas, kidney, and liver of all tested groups in this study. According to Sesterheim et al. [64], interleukin 1beta promotes the immune-mediated inflammatory response in T1DM by contributing to the autoimmune destruction of pancreatic beta cells. Dogan et al. [65] reported elevated levels of inflammatory markers, such as interleukin 1beta, in newly diagnosed T1DM patients, as well as in DM progression, implying that systemic inflammation may play a role in the disease’s development and complications. Indeed, cytokines such as TNF-α-, NO, interleukin 1beta, IL-6, and IL-10 levels are increased in response to acute liver injury caused by free radical production [66,67]. Significant reductions in interleukin 1beta levels after administration of *Ephedra foeminea* extract are attributed to *Ephedra foeminea’s* hepatoprotective properties, which might be due to its ability to inhibit proinflammatory cytokines such as interleukin 1beta.

Natural constituents, according to most studies, reduce oxidative stress inflammation and hence protect β cells [68]. In this work, *Ephedra foeminea* aqueous extract was administered orally to streptozotocin-induced diabetic rats and found to have antihyperglycemic and antioxidant activities. The enzyme activities of superoxide dismutase, catalase, and glutathione peroxidase were found to be increased in diabetic rats [69]. The increase in antioxidant enzyme levels shown in G2 could indicate an excess of reactive oxygen species production [70]. The extract of *Ephedra foeminea* appears to cause a significant decrease in glutathione peroxidase level, indicating that they play an antioxidant role by lowering the levels of reactive oxygen species. When STZ-induced diabetic rats were treated with limonene, their insulin production considerably increased [71]. According to our findings, limonene, one of the key chemicals included in *Ephedra foeminea* extract, can increase beta-cell regeneration by lowering oxidative stress caused by STZ, which can then boost insulin secretion, resulting in glucose homeostasis in diabetic rats. Thiamine administration reduces the degenerative abnormalities in the liver and kidneys associated with diabetes [72]. The presence of this important component, which has been found in *Ephedra foeminea* extract, may also play a role in diabetes control via vitamin involvement.

## 5. Conclusions

*Ephedra foeminea* has traditionally been used to treat a variety of ailments, including inflammation and cancer, but it was only recently noticed that it has antidiabetic activities. Our studies showed that *Ephedra foeminea* aqueous extract might markedly enhance glucose and lipid homeostasis in a diabetic rat model. In addition, *Ephedra foeminea* extract reduced oxidative stress, corrected lipid levels, and improved liver and kidney functions. Some limitations of the current study should be acknowledged, particularly the lack of results for ketone bodies estimation, OGTT, insulin estimation, and studying the pharmacokinetics of the *Ephedra foeminea* extract. All confounding factors that may occur because of the effect of insulin resistance and cell dysfunction, which infer the mechanism of the effect of the extracts, should be taken into consideration in our future studies. Future pharmacological and clinical studies are needed to determine the toxicity and protective effects of *Ephedra foeminea* extract on the liver and kidney, as well as to explain the extract’s specific mechanism(s) of action. It appears that *Ephedra foeminea* aqueous extract has the potential to treat inflammation and ROS-related abnormalities in diabetic patients.

## Figures and Tables

**Figure 1 nutrients-14-02338-f001:**
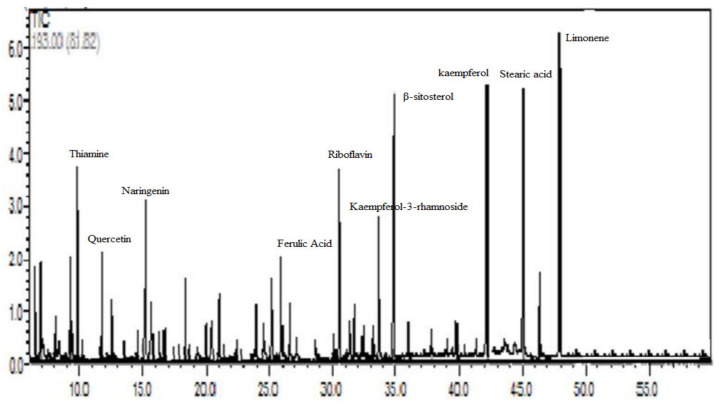
HPLC chromatogram of *Ephedra foeminea* aqueous extract eluted by 0.1% (*v*/*v*) formic acid in water and 0.1% (*v*/*v*) formic acid in acetonitrile.

**Figure 2 nutrients-14-02338-f002:**
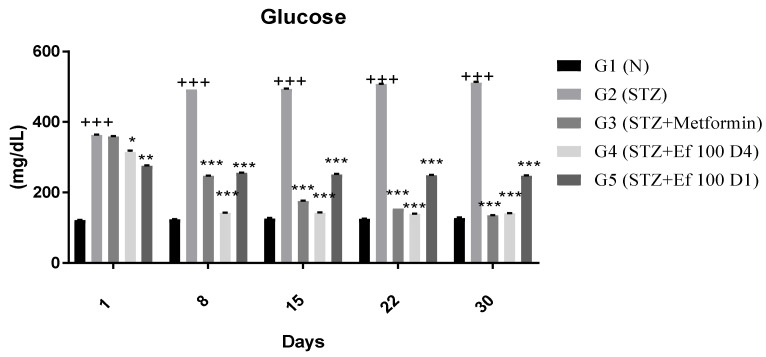
Effects of different treatments on blood glucose levels (mg/dL) in different experimental groups. Day 1: the day of the STZ injection, G1: normal, nondiabetic rats treated with normal saline solution orally, G2: diabetic rats treated with normal saline solution orally, G3: diabetic rats treated with metformin (100 mg/Kg) orally starting from Day 4, G4: diabetic rats treated with *Ephedra foeminea* extract (100 g/kg) orally starting from Day 4, G5: diabetic rats treated with *Ephedra foeminea* extract (100 g/kg) orally from Days 1 to 4. Bars represent the mean of the glucose concentration ± SD of 6 rats. Blood glucose levels were significantly decreased in induced diabetic rats treated with metformin (G3) and *Ephedra foeminea* extract (G4 and G5) compared to the untreated induced diabetic rats’ group (G2). The results are expressed as means ± SD (*n* = 3–4 dependent replicates). *: *p* < 0.05, **: *p* < 0.01, ***: *p* < 0.001 compared to G2: diabetic rats, +++: *p* < 0.001 compared G1: nondiabetic rats. The result were analyzed using one way ANOVA.

**Figure 3 nutrients-14-02338-f003:**
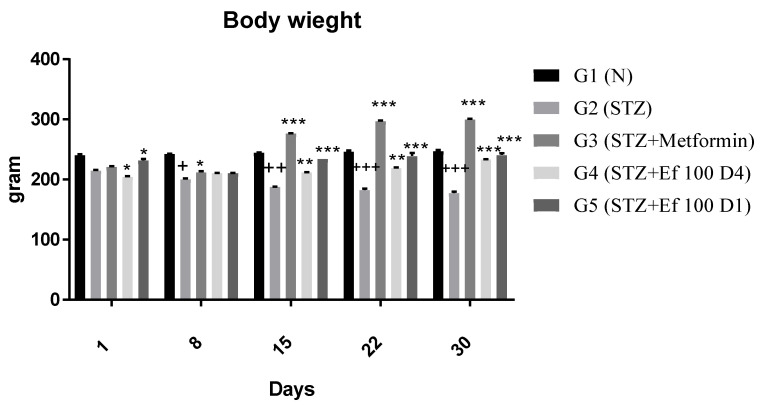
Effects of different treatments on body weight (g) of different experimental groups. Day 1: the day of the STZ injection, G1: normal, nondiabetic rats treated with normal saline solution orally, G2: diabetic rats treated with normal saline solution orally, G3: diabetic rats treated with metformin (100 mg/Kg) orally starting from Day 4, G4: diabetic rats treated with *Ephedra foeminea* extract (100 g/kg) orally starting from Day 4, G5: diabetic rats treated with *Ephedra foeminea* extract (100 g/kg) orally from Days 1 to 4. Bars represent the mean of the body weight ±SD of 6 rats. From Day 15, body weights were significantly (*** indicates *p* < 0.0001) increased for diabetic rats treated with metformin (G3) and *Ephedra foeminea* extract (G4 and G5) compared to the untreated induced diabetic rats (G2). The results are expressed as means ± SD (*n* = 3–4 dependent replicates). *: *p* < 0.05, **: *p* < 0.01, ***: *p* < 0.001 compared to G2: diabetic rats. +: *p* < 0.05, ++: *p* < 0.01, +++: *p* < 0.001 compared G1: nondiabetic rats. The result were analyzed using one way ANOVA.

**Figure 4 nutrients-14-02338-f004:**
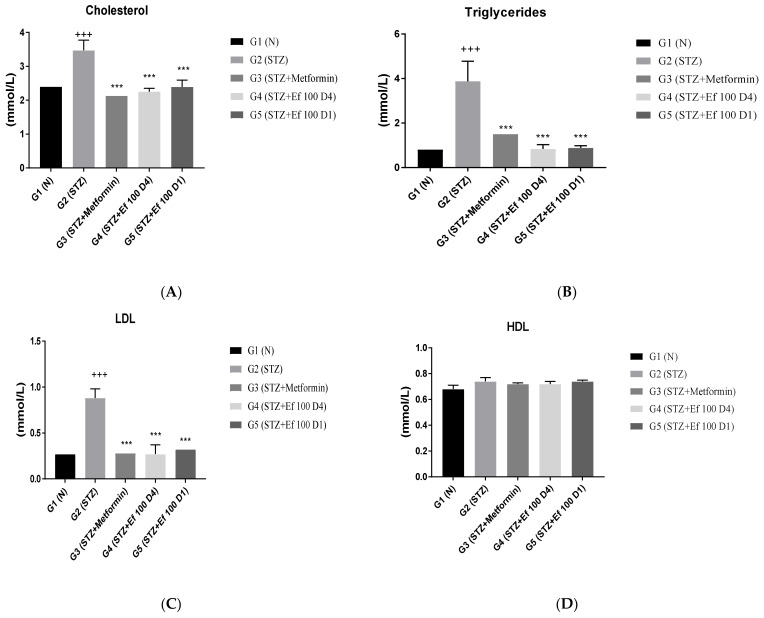
Effects of different treatments on serum lipid profiles in different groups. On Day 30, the concentrations of cholesterol (**A**), triglycerides (**B**), LDL (**C**), and HDL (**D**) were measured in all groups. Bars represent the mean of the parameters ± SD of 6 rats. Cholesterol, triglycerides, and LDL concentrations were significantly (*** *p* < 0.0001) lower in induced diabetic rats treated with metformin (G3) and *Ephedra foeminea* extract (G4 and G5) compared to the untreated induced diabetic rats (G2). The results are expressed as means ± SD (*n* = 3–4 dependent replicates), ***: *p* < 0.001 compared to G2: diabetic rats. +++: *p* < 0.001 compared G1: nondiabetic rats. The result were analyzed using one way ANOVA.

**Figure 5 nutrients-14-02338-f005:**
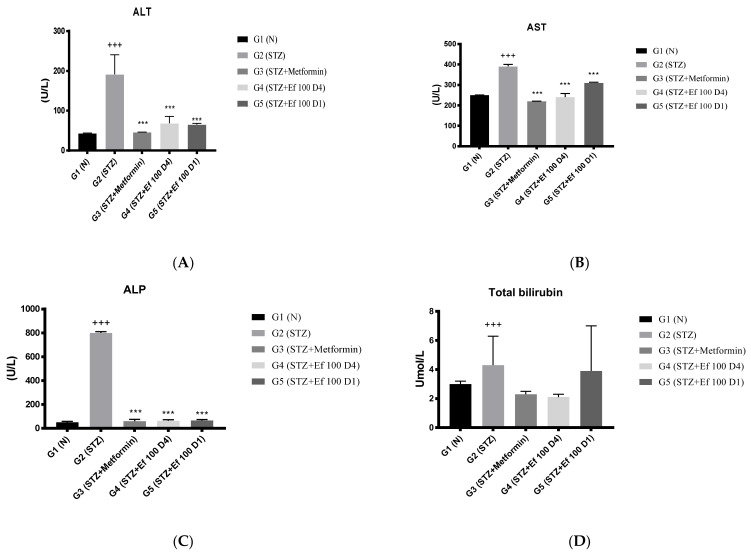
Effects of different treatments on serum markers of the liver in different experimental groups. On Day 30, the concentrations of ALT (**A**), AST (**B**), ALP (**C**), total bilirubin (**D**), and total protein (**E**) were measured in all the groups. Bars represent the mean of the parameters ± SD of 6 rats. The concentrations of ALT, AST, and ALP were significantly (*** *p* < 0.0001) decreased in induced diabetic rats treated with metformin (G3) and *Ephedra foeminea* extract (G4 and G5) compared to the untreated induced diabetic rats (G2). Concentrations of total bilirubin and total proteins did not differ significantly compared to the untreated induced diabetic rats (G2). The results are expressed as means ± SD (*n* = 3–4 dependent replicates), ***: *p* < 0.001 compared to G2: diabetic rats. +: *p* < 0.05, +++: *p* < 0.001 compared G1: nondiabetic rats. The result were analyzed using one way ANOVA.

**Figure 6 nutrients-14-02338-f006:**
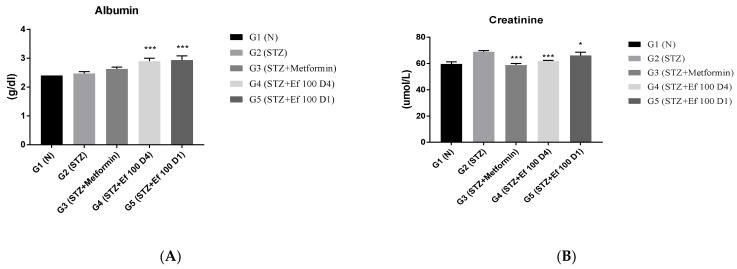
Effects of different treatments on serum markers of creatinine, albumin, urea, uric acid, and electrolytes in different experimental groups. On Day 30, the concentrations of albumin (**A**), creatinine (**B**), urea (**C**), uric acid (**D**), sodium (**E**), phosphorus (**F**), and chloride (**G**) were measured in all the groups. Bars represent the mean of the parameters ± SD of 6 rats. The concentrations of creatinine, urea, sodium, and ALP were significantly decreased in induced diabetic rats treated with metformin (G3) and *Ephedra foeminea* extract (G4 and G5) compared to the untreated induced diabetic rats (G2). However, the concentrations of albumin increased compared to the untreated induced diabetic rats (G2). The results are expressed as means ± SD (*n* = 3–4 dependent replicates). *: *p* < 0.05, **: *p* < 0.01, ***: *p* < 0.001 compared to G2: diabetic rats. +: *p* < 0.05 compared G1: nondiabetic rats. The result were analyzed using one way ANOVA.

**Figure 7 nutrients-14-02338-f007:**
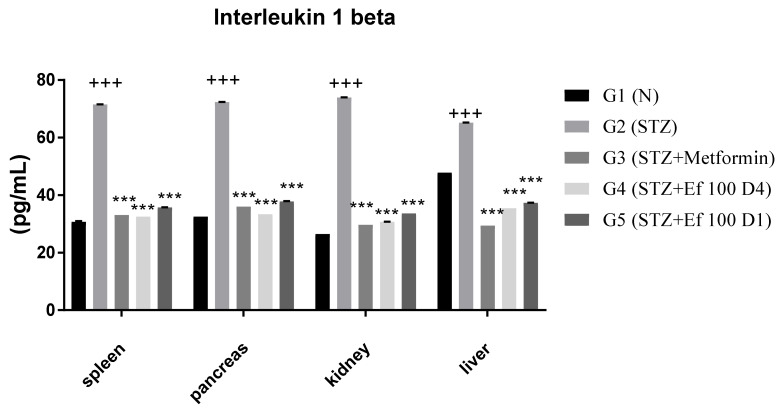
Concentrations of interleukin 1beta in the spleen, pancreas, kidney, and liver of all groups. Bars represent the mean of the interleukin 1beta concentration ± SD of 6 rats. On Day 30, the animals were anesthetized, their organs, including liver, spleen, pancreas, and kidney, were collected, and the concentrations of interleukin 1beta in these organs were measured. The interleukin 1beta concentrations were significantly (*p* < 0.0001) decreased in induced diabetic rats treated with metformin (G3) and *Ephedra foeminea* extract (G4 and G5) compared to the untreated induced diabetic rats’ group (G2). The results are expressed as means ± SD (*n* = 3–4 dependent replicates). ***: *p* < 0.001 compared to G2: diabetic rats. +++: *p* < 0.001 compared G1: nondiabetic rats. The result were analyzed using one way ANOVA.

**Figure 8 nutrients-14-02338-f008:**
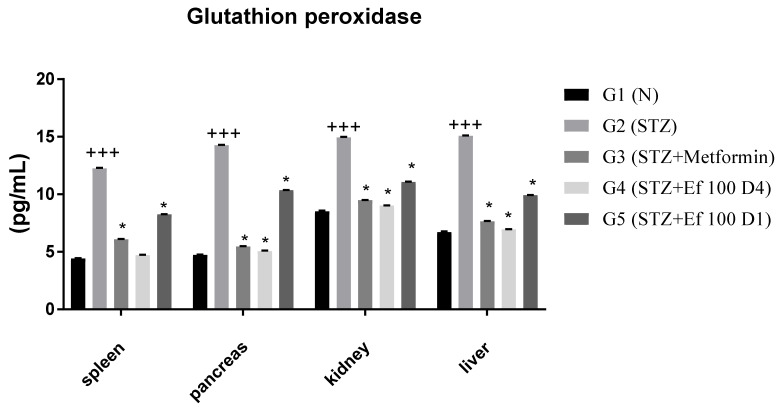
Concentrations of glutathione peroxidase in the spleen, pancreas, kidney, and liver of the experimental groups. Bars represent the mean of the glutathione peroxidase concentration ± SD of 6 rats. On Day 30, the animals were anesthetized and their organs, including liver, spleen, pancreas, and kidney, were collected and the concentrations of glutathione peroxidase in these organs were measured. The glutathione peroxidase concentrations were significantly (*p* < 0.05) decreased in induced diabetic rats treated with metformin (G3) and *Ephedra foeminea* extract (G4 and G5) compared to the untreated induced diabetic rats’ group (G2). The results are expressed as means ± SD (*n* = 3–4 dependent replicates). *: *p* < 0.05 compared to G2: diabetic rats. +++: *p* < 0.001 compared G1: nondiabetic rats. The result were analyzed using one way ANOVA.

**Table 1 nutrients-14-02338-t001:** Flavonoids and phenolic contents identified in *Ephedra foeminea* aqueous extract using LC-MS.

	Compound	MS Fragmentation Pattern	M.W	RT	%
**1**	Kaempferol	285, 213, 229	286.24	42	6.2
**2**	Naringenin	151, 177, 107	272.25	15.2	2.8
**3**	Kaempferol-3-rhamnoside	285, 431.1, 284.03, 255, 227	432.4	33.8	2.3
**4**	Quercetin	257, 229, 285	302.23	11.7	2.2
**5**	Ferulic acid	149, 134, 178	194.18	26	2
**6**	Epicatechin	123, 139, 165	290.27	15.3	1.6
**7**	ρ-Hydroxybenzoic acid	95.1, 121.1, 98	138.12	27	1.57
**8**	Luteolin	285, 241, 175	286.24	30.0	1.4
**9**	Catechin	139, 165, 123	290.27	31.3	1.35
**10**	Epigallocatechin	305.06, 306.07, 303.05, 304.05, 275.05	306.27	36	1.3
**11**	Quercetin-3-glucoside	465.1, 304.05, 301.03, 305.05	464.4	28.3	1.25
**12**	Gallocatechin	125, 137, 109, 139, 124	306.27	40.5	1.25
**3**	Hesperidin	301.07, 302.07, 286.04, 257.08, 325.07	610.6	41.3	1.25
**14**	Epiafzelechin	97.02, 273.07, 205.08, 137.02, 189.05	274.27	39.2	1.25
**15**	Vitexin	415.1, 397.1, 367.1	432.4	23	1.2
**16**	Herbacetin	303, 169, 257	302.23	13.5	1.19
**17**	Luteolin-7-glucoside	447.09, 285.04, 284.02, 284.05, 447.2	448.4	10.1	1.15
**18**	Isovitexin	415.1, 367.1, 313	432.4	17.5	1.1
	Total				32.36%

**Table 2 nutrients-14-02338-t002:** Terpenes, organic acids and other contents identified in *Ephedra foeminea* aqueous extract using LC-MS.

	Compound	MS Fragmentation Pattern	M.W	RT	%
**1**	Limonene	93.9, 68.6, 136.6, 121.3, 67.3	136.23	48	6.3
**2**	Stearic acid	265.4, 283.4, 266.5	284.5	45	5.9
**3**	β-Sitosterol	43, 414, 41, 55, 57, 107	414.7	35	5.5
**4**	Thiamine	263.1, 233.2, 147.1, 171.1, 58.9	265.36	10	4.1
**5**	Riboflavin	243, 359, 282	376.4	31.5	3.1
**6**	Quinaldic acid	174.05, 128.04, 175.05, 156.04	173.17	9.5	1.74
**7**	Kynurenic acid	171, 190, 144	189.17	7	1.7
**8**	Benzoic acid	79.05, 123.04, 105.03, 77.03, 106.03	122.12	6.5	1.7
**9**	Hexadecanoic acid	237.3, 255.3, 227.1	256.42	25.3	1.7
**10**	γ-Terpinene	93, 91, 121, 136, 77, 79	136.23	46.5	1.65
**11**	Vanillic acid	125, 93.1, 151.1	168.15	21	1.64
**12**	D-Norpseudoephedrine	134, 117, 115, 91, 119	151.21	13.8	1.6
**13**	Trans-aconitic acid	157.1, 133.1, 143.1	174.11	24	1.59
**14**	Nonacosanol	43, 42, 44	424.8	32	1.57
**15**	Linolenic acid	279.23, 261.21, 243.21, 95.08, 81.06	278.4	8.5	1.54
**16**	Ascorbic acid	87, 69, 147, 113,85	176.12	20.5	1.5
**17**	Malic acid	115, 87.1, 71.1	134.09	20	1.5
**18**	Fumaric acid	98.9, 45.68, 99.39, 116.35, 52.33	116.07	24.4	1.5
**19**	Oxalic acid	69.9, 68.23, 89.49	90.03	16.7	1.45
**20**	ρ-Coumaric acid	147.1, 123.2, 121.2	164.16	16.2	1.45
**21**	Cinnamic acid	131.1, 131.9	148.16	14.5	1.42
**22**	Dibutyl phthalate	205, 149, 204	278.34	32.5	1.35
**23**	D-Pseudoephedrine	91.12, 91.5, 114.8, 132.1, 65.3	165.23	33.1	1.3
**24**	Epigallocatechin	305.06, 306.07, 303.05, 304.05, 275.05	306.27	36	1.3
**25**	Citric acid	175, 147.1, 170.1	192.12	37.9	1.3
**26**	Caffeic acid	135.1, 135.9	180.16	40.0	1.3
**27**	Chlorogenic acid	191.05, 192.05, 93, 173.04, 353.1	354.31	43.5	1.25
**28**	Linalool	71.04, 95.08, 81.07, 69.07, 57.07	154.25	28.5	1.25
**29**	γ-Eudesmol	189, 204, 161,59	222.36	39.2	1.25
**30**	L-Ephedrine	166, 148, 167	165.23	22.5	1.2
**31**	1,8-cineole	43	212.28	18.0	1.15
**32**	Niacin	106.1, 80	123.11	21.5	1.1
	Total				63.9%

## Data Availability

Study data are available from the authors upon request.

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
