# Peer review of "Antioxidant and Antihyperglycemic Effects of Ephedra foeminea Aqueous Extract in Streptozotocin-Induced Diabetic Rats"

_nutrients, 2022, doi:10.3390/nu14112338_

Round 1

Reviewer 1 Report

In this research, the authors report that Ephedra Foeminea extract has beneficial effect in streptozotocin-induced diabetic rats due to its antyhyperglycaemic, lipid lowering, antiaxidative and antiniflammatory action. The results show the effect of the extract is similar to the one observed after a well-establish antidiabetic oral medicine-metformin.

The authors identified the content of the extract, used multiple metabolic  and enzymatic pathways to prove their hypothesis and anaylzed the effect of the extract on several organs in rats. So the study is interesting and well-conducted.

However, there are a few issue that should be clarified and modified:

  1. line 69--antidiabetic activity - better property
  2. line-71- Did you analyze IL 1 level or activity???
  3. In methods : It is described how the aqueous extract was prepared. However, there is no linknig to any literature, pharmacological procedure or Pharmacopoiea sources. Is the method of the extract preparation widely used or just developed by the authors?
  4. Generally metods are described in details but there are no references for any of the method used.

Please refer to the literature. The same with the rats preparatin and diabetes induction.

  1. How do you know 100mg/kg of the extract is the most effective one? Why did you use such a concentration. Refer the dosage the the one people use in the alterantive medicine in Jordan or elsewhere.
  2. line 155- In your studies the extract 100mg/kg was used but then you reffer others used 4000-8000mg/kg. Why are such differences in the dosage?
  3. line 175- so was it level or activity of IL-1?
  4. line 231-why was glucose higher in group g5 than in g4?
  5. line 238- 100g/L or kg?
  6. why is sodium and phophorus reduced after the extract treatment?-line 321-322
  7. line 336- add literature to the statement. Is it only IL-1 a biomarker of DM risk? Why did you chose thiis one?
  8. line 354- was it activity or concentration of gluthatione peroxidase?
  9. Did you check any level of the extract in blood of the rats to determine the bioavailability. Please discuss this issue at least.
  10. Please compare meformin to the extract. Would the extract be more beneficial than metformin?

Author Response

Dear editor:

It is a pleasure to re-submitting the revised manuscript (no # 1708160) entitled " Antioxidant and Antihyperglycemic Effects of Ephedra Foeminea Aqueous Extract in Streptozotocin-Induced Diabetic Rats" for consideration to be published in the nutrients.  We have found the comments provided by the reviewer to be valuable and have improved the manuscript.  We believe that the manuscript is currently more suitable for publication in its revised format. 

Reviewer 2 Report

The manuscript (nutrients-1708160) reported the antioxidant and antihyperglycaemic effects of Ephedra foeminea aqueous extract in streptozotocin-induced diabetic rats. Besides the extensive work done, the study lacks substantially in several aspects such, as background-rationale, methodology, result presentation-interpretation, writing, and language.

The following comments could help the authors to improve the manuscript:

  1. Please write the botanical names always in italics.
  2. The second paragraph gives the background of herbal medicine and its use in several diseases, however, 3rd paragraph vaguely says- Many people claim that it is beneficial for treating cancer and possesses a wide range of biological activities, including anti-asthmatic, anti-inflammatory, antimicrobial, antiproliferative, and 64 hypoglycemic properties [11-13]. While references 11-13 don't have any information about the hypoglycemic potentials of Ephedra Foeminea. 
  3. The introduction doesn't give any background or rationale behind testing the antihyperglycemic effects of Ephedra foeaminea aqueous extract in Streptozotocin-Induced Diabetic rats.
  4. Materials and methods: Plant material should be authenticated by certified/qualified botanists.
  5. There should be a reported ethnopharmacological basis for using aqueous extract only. If it has traditional use, there must be some documented proof. However, in exploratory studies, mostly different kinds of extracts are used. Furthermore, not all the phytochemicals listed need to be water-soluble. If so, the details should be provided.
  6. What was the basis of the LC-MS method used? Is it standardized for the first time or reproduced from the literature. It should be mentioned in the manuscript.
  7. The antioxidant activity could have been more realistic using in vivo models, mainly the tissue samples from the experimental animals.
  8. What was the reason behind choosing male rats? However, diabetes is common in males and females.
  9. Section 2.4.2 says the control group was given citrate buffer, but the next section says that non-diabetic rats were only given saline solution. This needs to be clarified.
  10. The groups could be simply named Normal control; STZ; STZ+Vehicle; STZ+Metformin; STZ+E. foeminea extract, etc., to make it easy to understand.
  11. The sample size should be mentioned in the method section.
  12. What was the rationale behind using metformin as a standard probably? STZ-induced diabetes replicates type-1 diabetes, whereas metformin is not an insulin sensitizer.
  13. Results: The figure for antioxidant activity is missing.
  14. Glucose levels must have been supported with serum insulin and OGTT data. 
  15. It is interesting to see that there was no weight loss in the STZ group.
  16. G-2 is having significantly higher cholesterol, triglycerides, and LDL levels, but is not shown in the graphs. This data also surprising that STZ causes hyperlipidemia which was constantly reversed by the treatment and metformin.
  17. Similar to the previous comment, liver function tests results are wrongly shown. Ketoneure estimation could have been more meaningful here.
  18. IL-1 beta and glutathione levels are surprisingly lowered by the treatments in all the tissues.
  19. Discussion- Page 14, lines 397-400 suggest the hypoglycemic effects of metformin, however, it can be seen as a euglycemic effect. Similarly, performing is usually considered as safest oral hypoglycemic.
  20. The next sentence (line 401) says metformin causes weight gain, however, it contrasts the clinical viewpoint.

Author Response

(The authors gave the same response as above.)

Round 2

Reviewer 2 Report

The authors have answered several comments in the revised manuscript, but the justifications given are not appropriate in most places. 

In addition, a few major points are there which are unacceptable, such as- 

1- No weight gain was observed in healthy rats in 30 days.

2- Dose and route administration of diethyl ether (1.5 g/kg body weight, i.p.).

3- No information was given about the route of drug administration. 

4- There should always be a statistical comparison between the normal control, and disease groups, (G1 and G2).

A pointwise explanation of the author's reply is as follows.

Comment #1- Botanical names are always written as Ephedra foeminea, not Ephedra Foeminea.

Comment #2- If the citations (11-13) were errors, these should not be added to the manuscript. It represents false information. 

Reply to comment #3 is also a repetition of the previous one. However, the authors have contraindicated their point itself. Dra et al. 2019 have used methanolic fraction (not aqueous) to investigate the antioxidant activity. However, that was a different plant extract, different animal model, and in mice. Both the references provided are either non-relevant or misleading. None of them are related to Ephedra foeminea.

Comment #5- The statement and references used to justify the use of aq.  extract is not correct.

Comment #7- Biochemical estimation of glutathione peroxidase cant be represented as EXPRESSION. It is an enzyme, and measuring its activity would give an actual sense of its action.

Comments #9- Technically, the control group should be given Citrate buffer on day 0 when STZ was given and vehicle (used to dissolve extract) on the treatment days after day 4. STZ is always given in Citrate buffer.

What was the route of administration of extract and metformin?

Comment #10- The name/label of the grouping should indicate the treatments. Especially when someone looks at the graphs or tables, the, group name should give an idea of treatment.  Naming the in the following way will make it easy to follow, and it will be easy to see the treatments given to each group. STZ; STZ+Vehicle; STZ+Metformin; STZ+EFE 100, etc., 

Comment #14- The justification for not doing OGTT and insulin estimation is inappropriate. OGGT can be done by withdrawing the blood drop from the tail tip, and insulin could be measured in the same serum samples. Besides all the tests done, these tests are most relevant to estimating glucose homeostasis in rodents and patients.

Comment #15- Again, it was strange to see no weight gain in the control group in 30 days; however, some significant difference was shown on day one. It this practically impossible that non-diabetic rats didn't gain any weight in 30 days, while healthy rats could gain a few grams every day.

Comment #17- Ketone bodies can be easily estimated using biochemical methods. It is not a valid explanation.

Author Response

Dear Reviewer 2

Many thanks for your constructive points which we have responded to point by point. We hope that our answers are now satisfactory.

The authors have answered several comments in the revised manuscript, but the justifications given are not appropriate in most places.

In addition, a few major points are there which are unacceptable, such as-

1- No weight gain was observed in healthy rats in 30 days.

There was not significant increase in the body weight of healthy rats that could not be observed through the curve, which was 239 g at the start, 242 g after eight days, 244 g after 15 days, 47 g after 22 days, and 247 g at the end of 30 days. There is no "conclusive" evidence as to why the body weight of healthy rats in our experiments fluctuated slightly. However, similar results regarding the non-significant or remarkable body weight change in the control groups (Rats) compared to the streptozotocin-induced diabetic rats treated with aqueous extracts of other plants were also documented by others. Examples:

For healthy rats, there was no significant increase in body weight that could not be seen by the curve, which began at 239 g and increased to 247 g after 22 days before stabilizing at 30 days. However, no significant change in body weight was observed in the control groups (rats) when compared to streptozotocin-induced diabetic rats treated with aqueous extracts of other plants in another study (El-Ouady et al., 2020; Kalita et al., 2016).

1- El-Ouady, F., Lahrach, N., Ajebli, M., Haidani, A. E., & Eddouks, M. (2020). Antihyperglycemic effect of the aqueous extract of Foeniculum vulgare in normal and streptozotocin-induced diabetic rats. Cardiovascular & Haematological Disorders-Drug Targets (Formerly Current Drug Targets-Cardiovascular & Hematological Disorders)20(1), 54-63.

2- Kalita, H., Boruah, D. C., Deori, M., Hazarika, A., Sarma, R., Kumari, S., ... & Devi, R. (2016). Antidiabetic and antilipidemic effect of Musa balbisiana root extract: A potent agent for glucose homeostasis in streptozotocin-induced diabetic rat. Frontiers in pharmacology7, 102.

2- Dose and route administration of diethyl ether (1.5 g/kg body weight.).

The rats were anesthetized under general anesthesia induced by diethyl ether inhalation as described by Alsarayreh et al.

You are correct, yet there was a big mistake in the wording, since the correct version is (Diethyl ether inhalation was used to induce general anesthesia in the rats, as described by Alsarayreh et al. (2021)  

Alsarayreh AZ, Oran SA, ShakhanbehJM. Effect of Rhus coriaria L. methanolic fruit extract on wound healing in diabetic and non-diabetic rats. J CosmetDermatol. 2021;00:1–11. doi:10.1111/jocd.14668

3- No information was given about the route of drug administration.

The treatments were given orally

4- There should always be a statistical comparison between the normal control, and disease groups, (G1 and G2).

It’s been modified accordingly. A pointwise explanation of the author's reply is as follows.

Comment #1- Botanical names are always written as Ephedra foeminea, not Ephedra Foeminea.

It’s been modified accordingly

Comment #2- If the citations (11-13) were errors, these should not be added to the manuscript. It represents false information.

We agree that it is a reasonable comment, and hence these references have been removed.

Reply to comment #3 is also a repetition of the previous one. However, the authors have contraindicated their point itself. Dra et al. 2019 have used methanolic fraction (not aqueous) to investigate the antioxidant activity. However, that was a different plant extract, different animal model, and in mice. Both the references provided are either non-relevant or misleading. None of them are related to Ephedra foeminea.

Thank you for taking the time to read this. As a result, we can see that this statement is unnecessary, and the reference associated with it is likewise eliminated, as it contradicts what was previously stated (the use of the aqueous extract of Ephedra foeminea has not previously been mentioned).

Comment #5- The statement and references used to justify the use of aq.  extract is not correct.

Response: the statement and the reference used have been corrected.

Comment #7- Biochemical estimation of glutathione peroxidase cant be represented as EXPRESSION. It is an enzyme, and measuring its activity would give an actual sense of its action.

In our current experience, the concentration or level of the enzyme, rather than its activity, was measured using the glutathione peroxidase kit according to the manufacturer's instructions (Shanghai Coon Koon Biotech Co., Ltd) as shown in the link below, and this method is also  reported to measure the level of stress conditions (Stupin et al., 2017).

Stupin, A., Cosic, A., Novak, S., Vesel, M., Jukic, I., Popovic, B., ... & Drenjancevic, I. (2017). Reduced dietary selenium impairs vascular function by increasing oxidative stress in Sprague-Dawley rat aortas. International Journal of Environmental Research and Public Health, 14(6), 591.‏

Comments #9- Technically, the control group should be given Citrate buffer on day 0 when STZ was given and vehicle (used to dissolve extract) on the treatment days after day 4. STZ is always given in Citrate buffer.

Response: Thank you. This what was performed in this experiment, but it was not mentioned as most of the published articles are not stating that. However, in the methodology section of this study, this info has been added.

What was the route of administration of extract and metformin?

The treatments were given orally

Comment #10- The name/label of the grouping should indicate the treatments. Especially when someone looks at the graphs or tables, the, group name should give an idea of treatment.  Naming the in the following way will make it easy to follow, and it will be easy to see the treatments given to each group. STZ; STZ+Vehicle; STZ+Metformin; STZ+EFE 100, etc.,

Text has been modified accordingly

Comment #14- The justification for not doing OGTT and insulin estimation is inappropriate. OGGT can be done by withdrawing the blood drop from the tail tip, and insulin could be measured in the same serum samples. Besides all the tests done, these tests are most relevant to estimating glucose homeostasis in rodents and patients.

This sentence is added to the conclusion.

Some limitations of the current study should be acknowledged, particularly the lack of results for Ketone bodies estimation, OGTT, insulin estimation and studying the pharmacokinetics of the Ephedra foeminea  extract. All confounding factors that may occur as a result of the effect of insulin resistance, cell dysfunction, and infer the mechanism of effect of the extracts can be minimized by taking the results of these tests into consideration in the future.

Comment #15- Again, it was strange to see no weight gain in the control group in 30 days; however, some significant difference was shown on day one. It this practically impossible that non-diabetic rats didn't gain any weight in 30 days, while healthy rats could gain a few grams every day.  This is briefly added to the discussion

In fact, we agree with your viewpoint and thank you. As previously stated, there was a slight rise in body weight, and this explains in detail:

There was not significant increase in the body weight of healthy rats which could not be observed through the curve, which was 239 g at the start, 242 g after eight days, 244 g after 15 days, 247 g after 22 days, and 247 g at the end of 30 days. There was no "conclusive" evidence as to why the body weight of healthy rats in our experiments fluctuated slightly. However, similar results regarding the non-significant or remarkable body weight change in the control groups (Rats) compared to the streptozotocin-induced diabetic rats treated with aqueous extracts of other plants were also documented by others.

Comment #17- Ketone bodies can be easily estimated using biochemical methods. It is not a valid explanation.

This is added to the conclusion

Some limitations of the current study should be acknowledged, particularly the lack of results for Ketone bodies estimation, OGTT, insulin estimation and studying the pharmacokinetics of the Ephedra foeminea  extract. All confounding factors that may occur as a result of the effect of insulin resistance, cell dysfunction, and infer the mechanism of effect of the extracts can be minimized by taking the results of these tests into consideration in the future.
